# Heart Rate as a Correlate for the Emotional Processing of Body Stimuli in Anorexia Nervosa

**DOI:** 10.3390/bs13030215

**Published:** 2023-03-02

**Authors:** Stefanie Horndasch, Elisabeth Sharon, Anna Eichler, Holmer Graap, Gunther H. Moll, Oliver Kratz

**Affiliations:** 1Department of Child and Adolescent Mental Health, University of Erlangen-Nuremberg, 91054 Erlangen, Germany; 2Department of Psychosomatic Medicine and Psychotherapy, University of Erlangen-Nuremberg, 91054 Erlangen, Germany

**Keywords:** anorexia nervosa, adolescents, body stimuli, heart rate

## Abstract

In anorexia nervosa, aberrant emotional reactions toward body stimuli have been discussed. We investigated heart rate as a physiological marker when viewing body stimuli and hypothesized altered HR reactions toward those highly significant and emotional stimuli in anorexia nervosa. In total, 37 anorexia nervosa patients and 43 control participants viewed pictures of women of five different weight categories, while their cardiac activity was recorded. R-R intervals following picture onset were determined, and means were calculated for three distinct time periods. The overall change in HR relative to baseline across all picture categories was greater in the patient group than in the control group (significant effect of “group”, *p* = 0.002, partial η^2^ = 0.120). A significant decrease in HR 2 to 8 s after picture presentation was found for pictures of women of extreme weight in both participant groups (significant “category ∗ time segment interaction”, *p* = 0.01, partial η^2^ = 0.037) and correlated with scores of sociocultural attitudes toward the appearance for the extremely underweight category (r = −0.33, *p* = 0.005). Therefore, differential HR reactions for anorexia nervosa patients and control participants were found for body stimuli in general. The highest HR decelerations in response to pictures of strongly underweight and overweight women may reflect emotional processes such as anxiety due to social comparison.

## 1. Introduction

Excessive preoccupation with body size or shape is a diagnostic feature of eating disorders such as anorexia nervosa (AN). Increased attention toward body-related stimuli has been associated with a distorted interpretation of information concerning one’s own body and an increase in negative emotions (see, e.g., [1,2]). Heart rate (HR) is a valid physiological marker of emotional processes such as valence and arousal. Apart from HR accelerations in response to highly arousing stimuli, short-term HR decelerations occur with most emotional stimuli. In healthy participants, HR deceleration occurs to varying degrees in the first seconds after the presentation of emotional pictures, depending on the valence of the images: the deceleration is most pronounced when viewing unpleasant stimuli, moderate when viewing pleasant stimuli, and least pronounced when viewing neutral stimuli. The HR curves show a triphasic course when viewing pleasant and neutral stimuli (deceleration–acceleration–deceleration), whereas the curve shows more of a sustained deceleration when viewing unpleasant stimuli [3,4]. HR deceleration has previously been associated with attentional focus on gathering information from the environment [5], and HR responses to pleasant and unpleasant stimuli have been shown to be modulated by interoceptive awareness [6]. Therefore, attention toward and engagement with stimuli seems to influence those responses. 

Due to the high salience of food and body shape in patients with eating disorders, research on stimulus processing has been conducted on different levels [7]. Studies on physiological responses, such as HR, have mainly focused on food stimuli. In individuals with binge eating disorder, the HR was lower when viewing high calorie as opposed to low calorie images, suggesting that food images may have had a potentially soothing effect [8], whereas in restrained eaters, the HR response to food images was greater than seen in controls and interpreted as potentially elevated arousal [9]. A study in patients with bulimia nervosa (BN) used a negative mood induction procedure and measured HR in response to images of feared foods and participants‘ own bodies [10]. BN patients in a sad mood responded with a stronger initial HR deceleration to food images compared to subjects with restrained eating. The authors interpreted this initial HR deceleration as an increased orienting response toward food under stress or negative emotions in BN patients. HR modulation by interoceptive awareness has been shown to be altered in AN in terms of increased HR prediction signaling, even at low levels of arousal [11].

Regarding affective but not disease-specific stimuli, the HR was similarly higher for pleasant and unpleasant images than for neutral images in participants with self-reported restrained eating, binge eating, controls, and food-deprived participants, without any group differences [9]. The absence of group differences regarding affective stimuli was not only seen in those non-clinical groups but also in a study with participants with BN [12]. A recent review reports a decoupling between self-reported emotional responses and physiological responses to affective stimuli across eating disorders [7]. Using a task employing HR self-counting and objective measurement, a deficit in interoceptive awareness was observed for subjects suffering from AN both at rest and in an emotional context [13].

To our knowledge, few studies have been conducted analyzing HR responses to body stimuli in patients with eating disorders. Green and Ohrt examined HR changes and subjective ratings of healthy adult women when viewing body images categorized as normal weight, underweight, and overweight, as well as neutral control images [14]. The highest level of negative affect was reported for underweight images, and the mean HR was significantly lower when viewing the overweight and underweight images than the normal weight body images. Overduin et al. found greater HR acceleration toward body stimuli compared to food and neutral stimuli in both women with and without restrained eating but without a clinically diagnosed eating disorder [15]. A study by Reichel et al. investigated the responses to images of extremely emaciated bodies on websites idealizing extreme thinness in a group of adolescent and adult female AN patients and healthy comparison subjects [16]. In the AN group, there was less HR deceleration (i.e., the minimum HR value after image presentation) and less overall deceleration (i.e., the HR mean value over an interval of 10 s) in response to these images, potentially related to an overall altered physiological arousal in AN. The aforementioned study in BN patients, however, could not detect differential reactions toward body as opposed to food stimuli in individuals with BN following a negative mood induction [10]. Other studies also suggest blunted psychophysiological responses to emotional content in AN, e.g., attenuated startle reflex reactivity to fear-inducing material [17], but similar peripheral physiological responses to emotion-eliciting film clips as in healthy women [18]. Adolescent AN patients who performed a body-related task (mirror exposure) showed a dissociation between psychophysiological reactivity (no differences in skin conductance responses compared to typical developing girls) and subjective responses (more negative emotion ratings) [19]. Similarly, in patients with AN, when looking at their own and other body silhouettes, there was no correlation between another psychophysiological measure—the pupil psychosensory reflex—and self-reported emotional arousal. An observed difference in pupillary reactions toward their own body silhouettes was not specific to AN but also present in healthy controls, similarly as an effect of the weight category drawn by underweight shapes for pictures of their own silhouettes [20].

Therefore, in previous studies on HR and other psychophysiological responses to body stimuli, a heterogeneous picture emerges. Patient self-reporting may be biased by social desirability or diminished introspective abilities and lack of emotional awareness [7,21]. Higher alexithymic tendencies are observed in AN in general and specifically in adolescent patients [22]. Psychophysiological measurement methods, on the other hand, are more objective and allow a time-sensitive recording of mental events during reception. In the case of HR measurement, the subdivision into time periods after image presentation seems to be important to reflect the dynamic course (see, e.g., [16]). The present study aims to investigate whether the altered subjective evaluation of body-related stimuli is reflected in heart rate acceleration or deceleration as peripheral physiological measures and whether there are differences between adolescent and adult female patients with AN and healthy female participants. These potential differences could hint toward HR as a neurophysiological correlate or marker of stimulus processing in AN, and analyzing correlations with subjective measures of eating disorder pathology or attitudes toward sociocultural appearance ideals could further elucidate the mechanisms leading to such emotional alterations in stimulus processing. A particular focus of this study is on assessing differential responses to distinct stimulus classes representing different body shapes.

## 2. Materials and Methods

### 2.1. Participants

The participant group consisted of 80 female adolescents and adults. Inclusion criteria for the patient group were an AN diagnosis according to ICD-10 criteria (ICD-10: F50.0 or F50.1), diagnosed by an experienced (child) psychiatrist or psychologist using the “Kinder-DIPS” diagnostic interview [23] in adolescents and the Structured Diagnostic Interview for Mental Disorders [24] in adults, female sex, and an age ≥ 12 years (adolescent group < 18 years, adult group ≥ 18 years). Twelve patients were taking selective serotonin-uptake inhibitors, four patients neuroleptic medication, and one patient pregabaline. The disease duration ranged from 4 months to 25 years. A total of 32 patients suffered from one or more comorbidities (depressive disorder: 28, anxiety disorder: 6, obsessive–compulsive disorder: 2, post-traumatic stress disorder: 2, personality disorder: 1 participant). AN patients were recruited and tested in the first 2–3 weeks of their inpatient stay at the Department of Psychosomatics and Psychotherapy or in the Department of Child and Adolescent Mental Health. 

Exclusion criteria were current or previous psychiatric disorder for the control group, and neurological disease, psychotic disorder, use of benzodiazepines, obesity, intellectual disability (IQ < 85), and insufficient understanding of the German language for both groups. A clinical examination was performed in all participants, and no major arrhythmia—with the exception of bradycardia—was detected.

All subjects (and their parents, in the case of minors) were informed, in detail, about the study, agreed to participate, and provided written informed consent. Control participants were recruited through public announcements or personal relationships. All subjects received a small compensation for their participation in the study. 

### 2.2. Procedure

First, all participants completed a German version of the Eating Disorder Inventory-2, which assesses eating disorder psychopathology (EDI-2; short German version with eight subscales [25]). It showed sufficient internal consistencies of those subscales with Cronbach’s alpha ranging from 0.58 to 0.90 [26]. Further, the SATAQ-G, a German version of the Social Attitudes Towards Appearance Questionnaire, which consists of 16 items and maps the extent of sociocultural influences on body image (e.g., “I’ve felt pressure from TV or magazines to lose weight“, “In our society, fat people are regarded as unattractive”), was used. High SATAQ scores reflect a strong endorsement of Western appearance ideals. Its three factors, pressure, awareness, and internalization, showed good internal consistency; it seems to be a satisfactory measure of sociocultural influences on body image [27].

The participants were seated comfortably 60 cm in front of a computer monitor (resolution 1024 × 768 pixels). The pictures were presented in random order for 8 s. After viewing each picture, participants were asked to rate the weight and attractiveness of the woman shown in the photograph. 

During picture viewing, ECG activity was recorded using two self-adhesive disposable Ag/Ag-Cl electrodes with hypertonic electrolyte gel on the left and right parasternal sides about 3 cm below the clavicle. A reference electrode was placed on the left arm. A signal amplifier and Brain Vision Recorder Software (both Brain Products GmbH, Gilching, Germany) were used to record cardiac activity. 

At the end of the experiment, body height and weight were measured, and the respective BMI value and BMI age percentile for adolescents [28] were calculated. 

### 2.3. Stimulus Material

Greyscale photographic stimuli were used showing women’s bodies in underwear in four positions (front, rear, profile standing, and profile sitting). The stimulus set is naturalistic and, at the same time, highly standardized (see Figure 1). Eight pictures from each BMI category were shown: extremely underweight (BMI 13.5–15.0 kg/m^2^), underweight (BMI 17.0–18.0 kg/m^2^), normal weight (BMI 20.0–22.5 kg/m^2^), overweight (BMI 25.0–30.0 kg/m^2^), and extremely overweight (BMI 50.0–65.0 kg/m^2^) [29]. Size assessment and valence of the pictures have been collected in a larger sample [1].

### 2.4. Data Analysis

Participants’ ages and psychopathology scores (EDI-2 and SATAQ-G) were analyzed using univariate analyses of variance (ANOVA) regarding differences between age groups (adults vs. adolescents) and participant groups (AN patients (“AN”) vs. control participants (“CO”)). BMI values were used for adults as opposed to BMI percentiles for adolescents; therefore, analyses were conducted separately for each age group. The significance level was adjusted after Bonferroni correction for multiple testing. 

HR data were processed and analyzed using BrainVision Analyzer software, v2.1 (Brain Products, Gilching, Germany). Raw ECG data were filtered (lower cutoff 40 Hz, upper cutoff 120 Hz, notch filter 50 Hz). Semi-automatic R peak detection was then performed and visually checked for accuracy and manually corrected if necessary. The time between two R peaks (R-R intervals) during the 8 s interval following picture onset and a baseline covering a period of 4 s before image presentation were then transformed into second-by-second heart rate (HR) in beats per minute (bpm) using weighted averages. Individual time segments were defined as 0–4 s, 2–6 s, and 4–8 s after image presentation. We decided on using 4 s time windows to determine the average heart rate because changes in HR due to emotion are typically slow, especially when HR variability is high, and a relatively long measurement period is required to capture them. Means were calculated for each picture category.

All statistical calculations were performed using the Statistical Package for Social Sciences (IBM SPSS) software, version 24.0. *T*-tests for independent samples were calculated for participant characteristics and questionnaire scores. 

HR difference scores to baseline values were calculated for each picture category and time segment, reflecting HR acceleration (positive values) or deceleration (negative values) in the respective time segment. In this way, general HR differences between groups (e.g., slower HR in a specific group) could be controlled for, and category-specific effects could be determined. Performance of the Kolmogroff–Smirnoff test for normal distribution for all HR measures showed *p*-values above 0.05 criterion in 80% of tests. After inspection of the histograms with normal distribution curves and Q-Q-Plots, HR across all picture categories was analyzed with a univariate ANOVA regarding differences between age groups and participant groups.

The difference scores for the separate picture categories were subjected to a repeated measures ANOVA with “BMI category” of the stimuli (extremely underweight, underweight, normal weight, overweight, and extremely overweight) and time segment (“1”: 0–4 s, “2”: 2–6 s, and “3”: 4–8 s after image presentation) as within-subject variables and between-subject factor “group” (AN patients (“AN”) vs. control participants (“CO”)). Greenhouse–Geisser correction was used when appropriate. For significant ANOVA results, post hoc *t*-tests were calculated. A significance level of *p* = 0.05 was set, which was adjusted after Bonferroni correction for multiple testing. Means (M) and standard deviations (SD) were reported. For time segments with significant HR changes, Pearson correlations with psychopathological scores (EDI-2, SATAQ-G) were calculated. 

## 3. Results

### 3.1. Sample Characteristics

The ages of the 80 female subjects in the study ranged from 12.3 to 49.0 years. The adolescents obtained significantly higher scores on the EDI-2 (total score) than adult participants. The participant groups (AN vs. CO) did not differ significantly with respect to age but with respect to BMI values/age percentiles, EDI-2 total scores, body dissatisfaction subscales, and SATAQ-G scores (see Table 1 and Table 2).

### 3.2. Heart Rate

#### 3.2.1. General HR Effects

The ANOVA of total HR across all picture categories revealed a significant effect of “age group” (generally lower HR in adults than in adolescents, F(1.76) = 9.52, *p* = 0.003, part η^2^ = 0.111) but no “participant group” effect (F = (1.76) = 0.59, *p* = 0.44, part η^2^ = 0.008) and “age group” × “participant group” interaction (F(1.76) = 1.76, *p* = 0.19, part η^2^ = 0.023). 

#### 3.2.2. Stimulus-Specific Effects

For changes in HR from baseline (HR difference scores reflecting acceleration or deceleration), the ANOVA revealed a medium effect of “group” and a significant “category × time segment” interaction of a small effect size (see Table 3). No significant three-way interaction was found.

The overall change in HR relative to baseline was greater in the patient group than in the control group (main “participant group” effect; for negative values reflecting HR deceleration, see Figure 2).

Post hoc tests for the “category × time segment” interaction revealed across both groups, all five picture categories, and after Bonferroni correction for multiple testing (α’ = 0.025), a non-significant increase in HR (t(79) = −2.00, *p* = 0.050) for time segment 2 vs. 1. For time segment 3 vs. 2, a significant decrease in HR (t(79) = 2.32, *p* = 0.023) can be observed. 

For a more detailed analysis of the significant “category × time segment” interaction, paired *t*-tests were computed: for each category, the first and second, as well as the second and third time segments, were compared. After Bonferroni correction for multiple testing (α’ = 0.005), significant results were obtained for the categories extremely underweight (t(79) = 3.31, *p* < 0.001) and extremely overweight (t(79) = 3.11, *p* = 0.003) for time segment 3 vs. 2. Thus, unlike the other categories, the HR responses for the extreme weight categories seem to occur predominantly 2 to 8 s after picture presentation as a significant decrease in HR, i.e., deceleration (see Figure 3).

For these decelerations (time segments 2 and 3), Pearson bivariate correlations with EDI-2 and SATAQ-G total scores were calculated. After Bonferroni correction for multiple testing (α’ = 0.005), a significant negative correlation between the SATAQ-G score and HR difference score for the category extremely underweight in time segment 2 (r = −0.33, *p* = 0.005) was obtained, signifying a greater HR deceleration in this segment for higher SATAQ-G scores. 

## 4. Discussion

Cue reactivity in eating disorders is a phenomenon of clinical importance that has not been fully elucidated to date. Our experimental design allowed us to use a more objective measure than purely subjective emotion ratings to observe responses to disease-specific images and differentiate between AN patients and healthy participants. 

### 4.1. General HR Effects

In adolescents, the HR throughout the experiment was faster than in adults. This is consistent with the literature on normal ranges of heart rate in children and adolescents and age-adjusted percentile curves [30]. However, the HR between AN patients and control participants did not differ in our study, which is contradictory to the HR measurements at rest and in an emotional situation [13] and a recent review reporting studies showing that patients with AN have a markedly lower HR than healthy control participants. These studies, however, generally used longer measurement intervals [31]. Other psychophysiological measures, such as skin conductance reactivity, were also reported to be blunted in AN under various conditions [19].

### 4.2. Participant Group Effects

A greater overall change in HR from baseline was seen in AN patients compared to control participants after the presentation of body images of different categories. The time course of HR changes (slight acceleration across the first two segments, then sustained deceleration) was similarly seen in a study by Bradley et al. for neutral and—with a greater deceleration in later time segments—pleasant emotional stimuli [3]. In contrast, for unpleasant picture stimuli, a sustained deceleration occurred in this study. 

In non-clinical populations reporting eating disorder symptoms as well as BN patients, emotional pictures without disease-specific content did not elicit differential HR responses [9,12]. The stronger overall deceleration observed only in the AN group in our study may be due to the fact that body images, in general, represent more unpleasant stimuli for AN patients than for the healthy control subjects, as seen in the emotion ratings of this and similar picture sets [1,32]. Explanations beyond the “unpleasantness” of the stimuli could be related to the specific nature of AN. The significance of the HR acceleration phase, i.e., in the present study, the second period in the triphasic course, is still unclear, but a greater initial HR deceleration and a less pronounced acceleration phase were observed for negative valence pictures (angry faces) compared to positive valence pictures (happy faces) and was interpreted as a stronger or longer-lasting attentional focus on aversive stimuli, which prevents the parasympathetic activity from subsiding [33]. In the context of the present study, this would imply an increased attentional bias toward body images in AN patients, possibly also influenced by altered interoceptive awareness, as shown in potentially anxiogenic situations—such as confrontation with body images—in AN [11]. Subjective ratings of the stimulus set have shown more negative ratings by AN patients with regard to specific body weight categories [1], while on a HR level, group differences did occur for all picture categories. Exposure toward their own bodies in a mirror did not elicit specific physiological reactions on a skin conductance level but more negative ratings on a subjective evaluation level in AN patients compared to healthy adolescents. It is therefore assumed that—possibly due to the insufficient integration of signals from the body and emotional experience—physiological phenomena and subjective experience are not always in line regarding situations that could be perceived as particularly “threatening”, such as emotional contexts in general [13], confrontation with one’s own body [18], body silhouettes [20] or—in our study—extremely overweight body shapes, for example. Contrary to our findings, studies using approach–avoidance paradigms could not—in spite of differences in the subjective ratings of stimuli—detect biased approach–avoidance reactions to thin and normal weight bodies in eating disorder vs. control participants [34,35] unless the bodies were depicted as the participants‘ own bodies. The HR reactions, therefore, seem to reflect not directly behaviorally measurable avoidance but more implicit emotional processes. 

### 4.3. Effects of Extreme Weight Categories

In our total sample, the strongest deceleration of HR was observed in the second and third time segments in response to stimuli showing extreme weight categories, namely, extremely underweight and overweight women. Similarly, a stronger HR deceleration was seen in healthy women toward underweight and overweight stimuli as opposed to normal weight stimuli [14]. Despite the different methodology used in the aforementioned study (the analysis of mean HR over an interval of 10 s vs. segmentation into time periods), the results suggest stronger HR responses to body shapes that deviate (in our study strongly) from normal weight. This could be seen in the context of the greater salience and novelty of “unusual” stimuli but also more specifically as a negative emotional reaction beyond the conscious level, comparable to the “fear bradycardia” described by [3], which has been demonstrated as a sustained HR deceleration to unpleasant stimuli. The reaction to overweight stimuli is also interesting in the context of an “anti-fat bias”, a phenomenon reflecting obese people and body shapes being associated with a wide range of negative characteristics on an explicit level but also reflected by implicit methods such as event- related potentials provoked by pictorial body stimuli [36]. This phenomenon is widely common in the general population from childhood onwards and could therefore play a role in stronger HR reactions to our overweight stimuli. Taking into consideration the study by Friederich et al., the effect could also be related—at least for the extremely underweight category—to anxiety due to social comparison processes [32] corresponding to reports of heightened pressure to achieve the societal appearance ideal, as reflected in correlations between the HR and SATAQ-G scores for this category in our total sample. There were no group differences regarding differential reactions to the specific weight categories. This may be in line with observations on the linguistic representation of bodies: women with AN showed no differences in biased cognitions about other bodies in general, but the experimental tasks revealed weight-based stereotypes both in patients and control participants [37]. Silhouettes of their own bodies also provoked pupil reactions specific to stimulus categories in AN and healthy participants [20]. 

### 4.4. Strengths and Limitations

The limitations of the study are the relatively small sample size and the low comparability with other studies due to differences in methodology (e.g., approaches to HR evaluation, stimuli, and the time period for their presentation). The sample was quite diverse in terms of comorbidities, disease duration, etc., and included participants taking medication; a sample with more “pure” characteristics would have been difficult to recruit. It consists of severely ill inpatients and female patients only, which may make it less representative of all AN patients. Depressive symptoms, which are common in AN, could have influenced HR reactions, albeit more in the direction of decreased reactivity [38]. The significance of HR in emotion research has not yet been fully clarified because the HR response to sometimes similar stimuli, e.g., fearful imagery, provoking cardiac acceleration and deceleration, reflects different affective–motivational processes, and no firm conclusions about, e.g., valence and arousal of our stimuli can be drawn. However, the results help to further elucidate the reflection of some ED symptoms and cognitive constructs as evaluated by subjective reports at a psychophysiological level. The inclusion of clinically diagnosed patients with AN in the study helps to differentiate disease-specific effects. A highly standardized picture set and analysis method highlight specific reactions to different body shapes.

## 5. Conclusions

In summary, the present study contributes to the research on emotional cue reactivity regarding body-related stimuli, such as those ever-present, e.g., in the media or comparison to peers, with pivotal significance and potentially harmful effects for AN patients. With the help of HR as a neurophysiological correlate, emotional alterations in stimulus processing in AN can be elucidated and related to psychological phenomena such as the internalization of sociocultural attitudes toward appearance. Stronger HR reactions to body stimuli underline the emotional significance in AN and point to the need for therapy tools targeting emotional aspects of body image and further research into the processing of disorder-specific content, e.g., in the media. Interventions promoting strategies to cope with emotional reactions when confronted with body images and tendencies to compare oneself with others (e.g., emotion recognition [39], mindfulness- and awareness-based therapies [40]) could reduce the negative impact of such situations on AN patients. Future research could focus on changes in such responses during/after such interventions and long-term treatment. 

## Figures and Tables

**Figure 1 behavsci-13-00215-f001:**
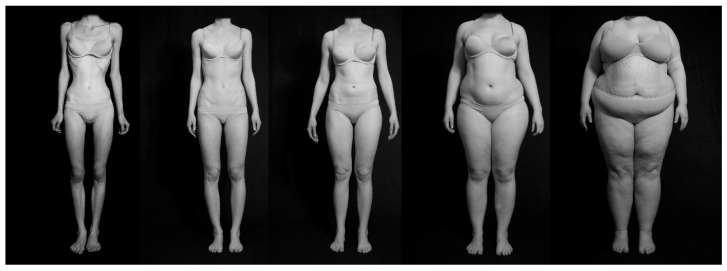
Stimuli of women’s bodies in five weight categories (from left to right: extremely underweight, underweight, normal weight, overweight, and extremely overweight).

**Figure 2 behavsci-13-00215-f002:**
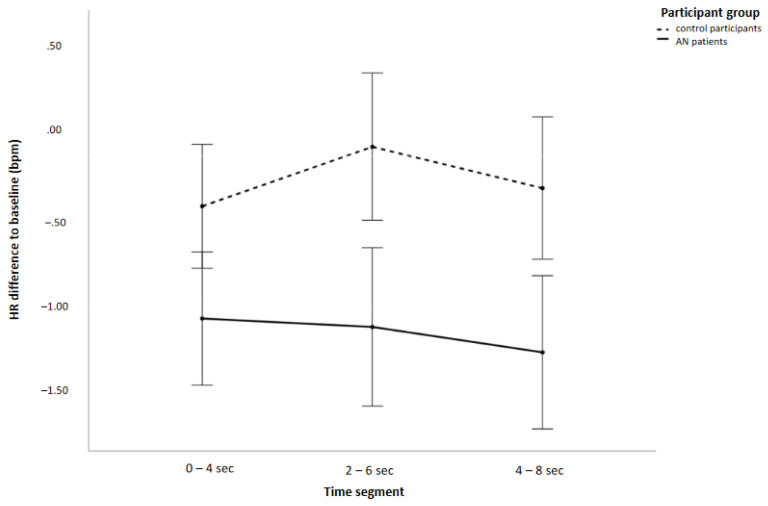
HR difference scores (bpm = beats per minute) across all picture categories for the 3 time segments and patient and control participants separately.

**Figure 3 behavsci-13-00215-f003:**
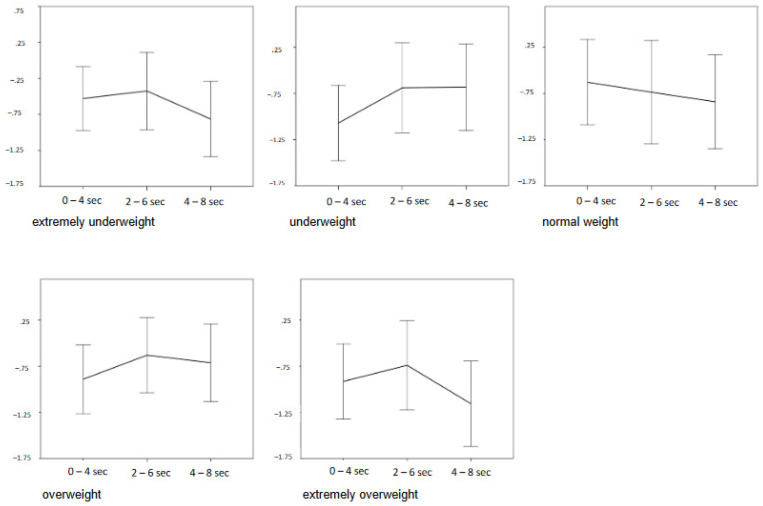
Time course of HR of all participant groups across time segments (*x* axis) in difference scores (*y* axis: mean HR in the respective time segment in relation to baseline HR, beats per minute), mean +/− 2 standard errors.

**Table 1 behavsci-13-00215-t001:** Overview of sample characteristics (AN = anorexia nervosa patients, CO = control participants, n = partial sample size, M = mean, SD = standard deviation, BMI = body mass index, EDI-2 = Eating Disorder Inventory-2, SATAQ-G = German version of the Social Attitudes Towards Appearance Questionnaire).

Age Group	Adolescents (n = 37)		Adults (n = 43)	
Participant Group	AN (n = 19)	CO (n = 18)	AN (n = 18)	CO (n = 25)
Age (years)	15.5 (1.9)	15.9 (1.5)	27.3 (7.9)	25.8 (7.1)
BMI (kg/m^2^)	15.9 (1.5)	20.8 (2.5)	16.5 (1.8)	21.8 (2.2)
BMI age percentile	5.5 (9.8)	52.4 (23.1)		
EDI-2 total score	277.8 (71.6)	205.6 (50.0)	347.7 (34.5)	213.1 (45.1)
EDI-2 body dissatisfaction	36.6 (11.6)	26.2 (9.4)	44.0 (8.0)	29.3 (8.7)
SATAQ-G total score	53.3 (21.4)	41.9 (11.5)	59.1 (12.1)	40.4 (10.2)

**Table 2 behavsci-13-00215-t002:** Results of univariate ANOVA regarding participant characteristics (* significant *p* values after Bonferroni correction (α’ < 0.01); BMI = body mass index, EDI-2 = Eating Disorder Inventory-2, SATAQ-G = German version of the Social Attitudes Towards Appearance Questionnaire).

Dependent Variable	Age Group Effect				Participant Group Effect				Age Group * Participant Group Interaction			
	Df	F	Partial η^2^	*p*	Df	F	Partial η^2^	*p*	Df	F	Partial η^2^	*p*
Age	1	79.31	0.511	<0.001 *	1	0.61	0.008	0.43	1	0.26	0.003	0.61
BMI (kg/m^2^)					1	58.56	0.613	<0.001 *				
BMI age percentile					1	63.07	0.612	<0.001 *				
SATAQ-G total score	1	0.43	0.006	0.52	1	20.85	0.227	<0.001 *	1	1.24	0.017	0.27
EDI-2 total score	1	9.37	0.131	0.003 *	1	66.74	0.518	<0.001 *	1	6.08	0.089	0.016
EDI-2 body dissatisfaction	1	5.51	0.073	0.022	1	32.04	0.314	<0.001 *		0.94	0.013	0.34

**Table 3 behavsci-13-00215-t003:** Results of the analysis of variance (ANOVA) between picture category, time segment, and group (* significant *p* values < 0.05).

Factor/Interaction	Df	F	Partial η^2^	*p*
Group	1	10.62	0.120	0.002 *
Category	4	0.42	0.005	0.78
Time segment	2	2.07	0.026	0.15
Group * time segment	2	2.14	0.027	0.14
Group * category	4	1.24	0.016	0.29
Category * time segment	8	2.96	0.037	0.010 *
Group * category * time segment	8	1.83	0.023	0.10

## Data Availability

Data will be made available by the authors upon request.

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
