# Peer review of "Heart Rate as a Correlate for the Emotional Processing of Body Stimuli in Anorexia Nervosa"

_behavsci, 2023, doi:10.3390/bs13030215_

Round 1

Reviewer 1 Report

The title is clear and direct, although long, it should be no more than 15 words.

The summary quickly sets out the basic content. The objective/hypothesis is missing at the beginning. Describes the methodology, extends in the results, lack of results with numerical data. Eliminate acronyms.

There are already works on this subject. The review carried out to justify the study is not current, include at least 50% of the references from the last five years.

The structure of the article and the arguments are logical and coherent. However, it is not explained why it is of interest to find out whether the evaluation of body-related stimuli is reflected in heart rate and whether there are differences between people with anorexia nervosa and the general population.

Clarify whether the diagnostic criteria are based on DSM-5 or ICD-11. Include reference to BMI categories.

The methodology seems appropriate and fits with the theoretical justification.

This is a small convenience sample. Add inclusion criteria, are there only women for any reason? The description of the validity and reliability of the instruments used is missing.

Data are analysed in relation to the objectives of the study.  Results are discussed in relation to the author's own studies and older published studies. Include other more recent studies and remove self-references.

The limitations reflected in the study reduce the importance of the topic of Heart Rate in emotion research. Justify the interest of the study.

Include a rationale for the research to make the conclusions clearer.

Include more current references from the last five years and eliminate self-references.

Correct errors in the text and in the list of references:

Some errors: in the text you should put the author(s) if there are two authors and the corresponding number (e.g. Green & Ohrt (12)). If there are more than two authors, put the first author et al. and the number (for example: Overduin et al. (13)). In indirect citations, put the reference number, not the author (missing in line 277) and the number and delete the year in line 283) and these authors must be in the list of references. On line 322 the author Bradley et al. (3) should appear. On line 330 put the number Friederich et al. (21).

Author Response

Thank you very much for your helpful comments. Please find our point-by-point responses as an attachment. 

Reviewer 2 Report

The manuscript reports an interesting study regards the physiological effects of picture presentation recorded with HR. The study has several interesting points, but also some aspects that need revision. Please, consider my points as suggestions for the implementation of the manuscript:

- I think the first line of the introduction should be deleted.

- Be sure to introduce all the acronyms before their use (see for example ED in line 44 or BN in line 52)

- I think the introduction would benefit from some more information about the interception in AN, due to the mutual effects of body reactions and cognitive concerns (see for example https://doi.org/10.1002/eat.22387 or  https://doi.org/10.1007/s40519-022-01394-7)

- have you evaluated the parametric distribution of the data? Because the subgroups are not so big.

- You recruited participants in the inpatient unit. Might the severity of these disorders have a role in the results?

- Did you randomize the pictures? Have you followed a specific order? This aspect is crucial and might produce some bias. 

- Does Figure 3 represent the HR of patients or controls? Adults or adolescents?

- have you collected data about the valence of the figures presented? Because you discussed this aspect in the paper but I am not able to find the data in the results.

- Moreover, I think your data should be discussed with the recent evidence regarding the role of implicit judgment in AN as regards body image because they are so related (see https://doi.org/10.1002/erv.2812).

Author Response

(The authors gave the same response as above.)

Round 2

Reviewer 1 Report

The manuscript has been improved, many suggestions have been corrected, but it still needs to be revised:

You need to include numerical results in the abstract

You have added more recent references. However, only 30% of the references are from the last five years.

There are errors in the references that have not been corrected. Correct them:

On line 330 (Peyser et al., 2020), delete the year and insert the number, add to in the list of references.

On line 337 Bradley et al. (2001), delete the year and insert the number [3].

On line 295 Friederich et al. (2006), delete the year and insert the number [28].

Author Response

Dear reviewer,

Thank you very much for the helpful comments.

Based on your suggestion, we have filled in the numerical results in the abstract.

We have added four more references from the last five years in order to include more recent evidence – with a focus on reactions to body silhouettes, alexithymia and the dissociation between subjective emotional evaluation and psychophysiological measures – in the introduction and discussion.

We have also the errors in the references that you pointed out and updated the reference list.

Reviewer 2 Report

The paper covers an interesting topic, and the authors have addressed all my concerns. Thus, I think it might be considered for publication.

Author Response

Dear reviewer, 

Thank you very much for your kind and interesting comments. We asked a colleague who is a native speaker to spell check the manuscript and corrected minor errors.